# Hierarchical Fusion for Online Multimodal Dialog Act Classification

**Md Messal Monem Miah**[*]    **Adarsh Pyarelal**[†]    **Ruihong Huang**[*]

[*]Texas A&M University, [†]University of Arizona

{messal.monem, huangrh}@tamu.edu, {adarsh}@arizona.edu

## Abstract

We propose a framework for online multimodal dialog act (DA) classification based on raw audio and ASR-generated transcriptions of current and past utterances. Existing multimodal DA classification approaches are limited by ineffective audio modeling and late-stage fusion. We showcase significant improvements in multimodal DA classification by integrating modalities at a more granular level and incorporating recent advancements in large language and audio models for audio feature extraction. We further investigate the effectiveness of self-attention and cross-attention mechanisms in modeling utterances and dialogs for DA classification. We achieve a substantial increase of 3 percentage points in the F1 score relative to current state-of-the-art models on two prominent DA classification datasets, MRDA and EMOTyDA.

## 1 Introduction

Dialog Acts (DAs), as described by Searle (1969), are the minimal units of linguistic communication that represent the speaker's intention behind an utterance in a conversation. Successful DA classification facilitates a range of downstream tasks including task-oriented dialog systems (Blache et al., 2020; Wang et al., 2020b), conversational agents (Ahmadvand et al., 2019; Wood et al., 2020), and dialog summarization (Oya and Carenini, 2014; Goo and Chen, 2018). Since DA classification is crucial for understanding spontaneous dialog (Stolcke et al., 2000), a significant amount of effort has been put into modeling DAs computationally. However, most of the research efforts on DA classification heavily rely on oracle transcriptions (Ortega and Vu, 2017; li et al., 2018; Li et al., 2019; Colombo et al., 2020; He et al., 2021), resulting in several limitations.

First, oracle transcriptions are typically inaccessible during real-time dialog processing, such as in conversational agent systems. These systems

| Speaker | Utterance | Dialog act |
|---------|-----------|------------|
| me018 | I guess when Sunil gets here he can do his last or something. | Suggestion |
| me013 | Yeah. | Acknowledgement |
| me013 | So we probably should wait for him to come before we do his. | Command |
| me006 | Okay | Backchannel |
| me018 | That's a good idea | Appreciation |

Table 1: Short segment of a dialog from MRDA corpus.

depend on automatic speech recognition (ASR) for transcriptions, which are often subject to noise and inaccuracies. Ortega et al. (2019) demonstrated that DA classification models trained on oracle transcriptions exhibit subpar performance when applied to noisy transcriptions.

Second, audio signals contain important acoustic and prosodic cues, which are crucial for DA classification (Jurafsky et al., 1998), but are overlooked when focusing solely on text transcriptions. An example of the importance of acoustic information for DA modeling can be found in the brief dialog excerpt presented in Table 1. The utterances labeled as *suggestion* and *command* are lexically very similar, making disambiguation based solely on textual content difficult.

In this research, we explore real-time DA classification, utilizing only the raw audio signals as input. However, the presence of noise in the audio signals and potential inaccuracies in the transcriptions generated by Automatic Speech Recognition (ASR) systems make the setting particularly challenging. This motivates us to investigate a multimodal framework that leverages the complementary nature of audio and textual representations, enabling reliable DA classification despite the limitations of noisy audio signals and imperfect ASR transcriptions. A few previous research efforts explore the setup but are limited by either poor dialog modeling or ineffective use of audio. He et al. (2018) make use

of ASR generated transcriptions and audio features but do not consider contextual information while modeling the utterances, resulting into drammatically lower performance in comparison to using oracle text.

We propose a novel multimodal approach that uses a hierarchical fusion technique to combine the two modalities at early-stage, resulting in superior performance compared to utilizing individual modalities for online DA classification. Our proposed approach leverages multimodal fusion strategies to effectively model both the utterances and their context. Additionally, we propose a state-of-the-art audio feature extraction process that enhances audio-based modeling of dialogs, leading to improved performance in DA classification. We release the codes for feature extraction and experiments in this repository.

## 2  Related Work

**Dialog Act Classification**   Several studies have explored different approaches to facilitate DA classification, with a focus on employing various neural network architectures and attention mechanisms for leveraging contextual information. Ortega and Vu (2017) use CNNs to acquire utterance representations and further investigate various forms of attention mechanisms to infuse context awareness into the utterance representations. Bothe et al. (2018) propose using a pre-trained character-level language model to generate sentence representations and a RNN to learn dialogue act from the current utterance representation and the context of previous utterances. li et al. (2018) introduce CRF-attentive structured network with integrated structured attention to simultaneously model contextual utterances and their corresponding DAs. Li et al. (2019) jointly model DAs and topics using a shared bidirectional GRU-based utterance encoder and task-specific attention mechanisms.

Raheja and Tetreault (2019) propose a hierarchical RNN with context-aware self attention to model various levels of utterance and dialog act semantics. Wang et al. (2020a) propose Heterogeneous User History (HUH) graph convolution network, complemented by denoising mechanisms in order to effectively integrate user's historical answers grouped by DA labels to improve DA classification. He et al. (2021) propose a speaker turn-aware approach where the turn embedding vector, learned based on the speaker label is combined with the utterance vector for further processing by an RNN to capture contextual information. Żelasko et al. (2021) explore the effects of broader context, capitalization and punctuation on DA classification and report strong result using XLNet and Longformer.

**Multimodality for Dialog Modeling**   Multimodality is widely explored in multimodal sentiment analysis (MSA) and emotion recognition in conversation (ERC) tasks. Hazarika et al. (2018) propose Conversational Memory Network (CMN) for ERC that uses GRU to model the utterances from fused feature representations and use attention mechanism to implement the memory network. Poria et al. (2017) present (i) CAT-LSTM, a contextual attention-based LSTM network for modeling utterance relationships and (ii) AT-Fusion, an attention-based fusion mechanism that enhances multimodal fusion for MSA. Tsai et al. (2019) introduce the cross- attention based Multimodal Transformer (MulT), an end-to-end model that expands upon the standard transformer network (Vaswani et al., 2017) to enable learning representations directly from unaligned multimodal streams for ERC.

However, unlike ERC and MSA, multimodality is less explored for DA classification. He et al. (2018) combine CNN for audio feature augmentation and RNN to model utterances from individual modalities and fuse them through concatenation. While Ortega and Thang Vu (2018) utilize CNN to capture intra-utterance context for both audio and textual modalities, they only consider dialog-level context for modeling textual representation. Saha et al. (2020) propose a Triplet Attention Subnetwork, incorporating self and cross attention to jointly model emotion and DA in dialogs.

## 3  Proposed Method

In this section, we will describe the components of the proposed model in detail (see Figure 1)

### 3.1  Task Description

A dialog consists of $n$ utterances ($U$) accompanied by their respective labels ($Y$), arranged in chronological order. Each utterance comprises a text component ($x_i^t$) and an audio component ($x_i^a$). Mathematically, a dialog with $n$ utterances can be represented as:

$$\{U, Y\} = \left\{ \left( x_i = \langle x_i^t, x_i^a \rangle, y_i \right) | i \in [1, n] \right\}. \quad (1)$$

Here, $x_i$ denotes the $i^{\text{th}}$ utterance, where $x_i^t$ represents the text component and $x_i^a$ represents the

audio component. A DA label $y_i$ from a predefined set of labels ($C$), is assigned to each utterance. Our proposed network takes this input data and aims to accurately classify the DA associated with each utterance. Since, we are interested in online DA classification, we only consider the raw speech signals as the input. We use Whisper (Radford et al., 2022) to obtain transcriptions.

## 3.2 Multimodal Feature Extraction

**Textual Features** We use a pre-trained RoBERTa (Liu et al., 2019) model to extract textual features. The textual representation from each utterance $x_i^t$ is passed through the pre-trained RoBERTa model, and the representations from the last hidden layer are extracted as the token embeddings. Mathematically $x_i^t$ can be represented as follows:

$$x_i^t = \{w_i^1, w_i^2, ..., w_i^k\} \in \mathbb{R}^{k \times d_t}. \quad (2)$$

Here, $w_i^j$ represents the $j^{\text{th}}$ token of the $i^{\text{th}}$ utterance, $k$ the total number of tokens in an utterance, and $d_t$ the embedding dimension.

**Audio Features** We use a pre-trained Whisper model to extract audio features. The audio signal corresponding with each utterance $x_i^a$ is passed through a pre-trained Whisper model and the hidden states from the last encoder layer are extracted as the frame embeddings. Mathematically $x_a^t$ can be represented as follows:

$$x_a^t = \{f_i^1, f_i^2, ..., f_i^m\} \in \mathbb{R}^{m \times d_a}. \quad (3)$$

Here, $f_i^j$ represents the $j^{\text{th}}$ frame of the $i^{\text{th}}$ utterance, $m$ the total number of frames in an utterance, and $d_a$ the embedding dimension.

The choice of Whisper as the audio feature extractor is motivated by two key factors. First, given the real-time nature of online DA classification, the usage of Whisper, which is already used for transcriptions, eliminates the necessity for a separate feature extraction model. This not only reduces the time requirements but also simplifies the inference process by leveraging the same pre-trained model. Second, we observe that using features extracted by Whisper leads to better DA classification results than using other popularly used features for similar tasks such as MFCC, eGeMAPS (Eyben et al., 2016), and even transformer-based features like Wav2Vec2 (Baevski et al., 2020) embeddings. We provide a detailed comparison of the model performance using these features in § 5.

## 3.3 Model Architecture

The proposed model consists of three primary components: (1) an *early-stage fusion encoder* that fuses the two modalities at the utterance level and incorporates attention-based dialog modeling using the fused utterances, (2) *unimodal encoders* that generate context-aware representations from each modality individually, and (3) a *late-stage fusion classifier* that combines the context-aware representations from both the early-stage fusion encoder and the unimodal encoders and applies linear layers to perform the final DA classification.

### 3.3.1 Early-stage Fusion Encoder

In this branch of the network, every utterance $x_i = \langle x_i^t, x_i^a \rangle$ within the dialog is processed by two Bi-LSTM networks. Specifically, Bi-LSTM$_t$ is used to model the textual utterance representation extracted from $x_i^t$, while Bi-LSTM$_a$ is used to model the audio utterance representation extracted from $x_i^a$. At any time $j$, the forward $\overrightarrow{\text{LSTM}_t}$ computes the forward hidden vector $\overrightarrow{h}_i^{j,t}$ based on the previous hidden vector $\overrightarrow{h}_i^{j-1,t}$ and the input token embedding, $(\mathbf{w}_i^j)$, while the backward $\overleftarrow{\text{LSTM}_t}$ computes the backward hidden vector $\overleftarrow{h}_i^{j,t}$ based on the opposite previous hidden vector $\overleftarrow{h}_i^{j+1,t}$ and the input token embedding $(\mathbf{w}_i^j)$. Subsequently, the forward and the backward hidden vectors are concatenated into the final hidden vector, $h_i^j$ of the Bi-LSTM model. Here, $h_i^j$ is a $2d_h$-dimensional vector where $d_h$ is the hidden dimension of the LSTMs. Mathematically,

$$\overrightarrow{h}_i^{j,t} = \overrightarrow{\text{LSTM}_t}(\mathbf{w}_i^j, \overrightarrow{h}_i^{j-1,t}) \quad (1)$$

$$\overleftarrow{h}_i^{j,t} = \overleftarrow{\text{LSTM}_t}(\mathbf{w}_i^j, \overleftarrow{h}_i^{j+1,t}) \quad (2)$$

$$h_i^{j,t} = \text{concat}(\overrightarrow{h}_i^{j,t}, \overleftarrow{h}_i^{j,t}) \in \mathbb{R}^{2d_h} \quad (3)$$

We use the self-attention mechanism to calculate the utterance embedding $s_i^t$ from the token embeddings as a weighted sum of all the hidden states from the Bi-LSTM network. To achieve this, the hidden vector $h_i^{j,t}$ is passed through a simple multi-layer perceptron (MLP) with a non-linearity function (tanh) to generate a new hidden representation. This, in turn, is passed through another single-neuron MLP followed by a softmax to determine the attention weight for each token, $\alpha_i^{j,t}$ in the utterance. This process can be represented by the following

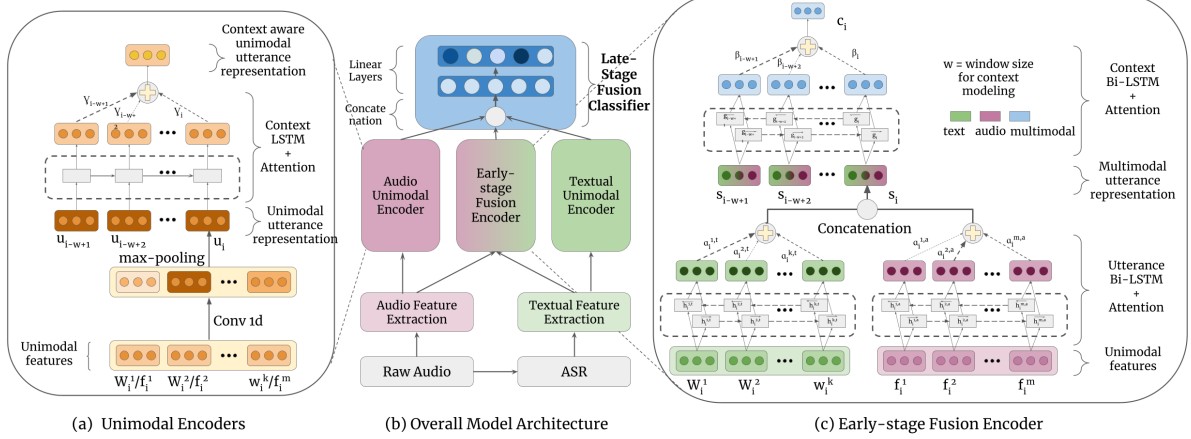

Figure 1: The proposed model architecture.

set of equations:

$$r_i^{j,t} = \tanh(W_1 h_i^{j,t} + b_1))) \qquad (4)$$

$$\alpha_i^{j,t} = \text{softmax}_j(W_2 r_i^{j,t}) \qquad (5)$$

$$s_i^t = \sum_j \alpha_i^{j,t} h_i^{j,t} \in \mathbb{R}^{2d_h} \qquad (6)$$

where $W_{i=1,2}$ and $b_1$ represent the weights and bias of the MLP layers respectively.

The audio utterance representation $s_i^a$ is computed by the Bi-LSTM$_a$ followed by self-attention mechanism in an identical manner. Finally the two representations are concatenated to compute the *multimodal utterance representation $s_i$*:

$$s_i = \text{concat}(s_i^t, s_i^a) \in \mathbb{R}^{4d_h} \qquad (7)$$

As an alternative design for early-stage fusion encoding, we investigate the integration of cross-attention between token and frame representations when generating the multimodal utterance representation. The objective of incorporating cross-attention is to learn a soft alignment score that captures the relationship between the textual and audio modalities within the same utterance. While calculating cross attention, we use either the text modality as source and audio as target or vice versa. Two cross-attention based utterance embedding are computed, one is $s_i^{t,a}$ where, the attention weights are calculated using the dot product of the two modality representations, $h_i^{j,t}$ and $h_i^{j,a}$ and based on the attention weights the token embeddings are aggregated to an utterance embedding. The $s_i^{a,t}$ is calculated in a similar manner. To achieve this, the token embeddings and frame embeddings are passed through individual simple MLP layers followed by a tanh activation to generate intermediate token or frame representations.

$$p_i^{j,t} = \tanh(W_3 h_i^{j,t} + b_3) \qquad (8)$$

$$p_i^{j,a} = \tanh(W_4 h_i^{j,a} + b_4) \qquad (9)$$

Pairwise dot-products between intermediate token and frame representations are computed using matrix multiplication. Considering, $P_i^t \in \mathbb{R}^{k \times 2d_h}$ and $P_i^a \in \mathbb{R}^{m \times 2d_h}$ are the two matrices of the intermediate token and frame representations of all the utterances, The following matrix multiplication is employed to compute the attention weights.

$$S = P_i^t \times (P_i^a)^T \in \mathbb{R}^{k \times m} \qquad (10)$$

Here, $k$ and $m$ represents the number of tokens in each utterance and the number of frames in each utterance, respectively, and $2d_h$ is the embedding dimension. Another MLP transformation is applied on the audio frame embedding matrix and finally multiplied with the with the attention weights to calculate the attended token embeddings,

$$P_i^{t'} = S \times MLP(P_i^a) \in \mathbb{R}^{k \times 2d_h} \qquad (11)$$

Finally, the average of all the cross-attended token embeddings per utterance is computed to generate the utterance embedding, $s_i^{t,a}$. Another cross-attended utterance embedding $s_i^{a,t}$ is also calculated in a similar manner. The final utterance representation is the concatenation of the self and cross attended utterance representations.

$$s_i = concat(s_i^t, s_i^a, s_i^{t,a}, s_i^{a,t}) \in \mathbb{R}^{8d_h} \qquad (12)$$

This utterance representation is then passed through a single-layer MLP that projects it to a lower-dimensional space for further processing.

Once we have generated the multimodal utterance representations, we incorporate contextual information from the previous utterances into the utterance representation. To achieve this, we use a window of width $w$ that includes the current utterance representation and $w$-$1$ previous utterance representations for modeling the context-aware utterance representation. The context modeling process follows a similar approach to the utterance modeling. The context encoder consists of a Bi-LSTM and attention mechanism. The Bi-LSTM encodes the utterance representations into hidden states $g_i$, while the attention mechanism calculates the attention weight $\beta_i$ for each of the hidden states. The final context-aware utterance representation is calculated as follows:

$$c_i = \sum_w \beta_i g_i \qquad (13)$$

The context-aware utterance representation is finally passed through a single-layer MLP that projects it to a lower-dimensional space.

### 3.3.2 Unimodal Encoders

While the early-stage fusion encoder focuses on fusing modalities effectively, our approach also incorporates two modality-specific encoders that model context-aware utterance representations from text and audio individually. To develop these unimodal encoders, we augment the model proposed by Ortega and Thang Vu (2018).

First, we use a 1D CNN followed by max-pooling to effectively model utterance representations from the token or frame vectors of an utterance. The kernel size and number of filters for CNN are two hyperparameters that we tune for the best performance of the model. Finally, we contextualize the utterance representations in a similar manner as described in the context modeling for early-stage fusion. To achieve this, we use an LSTM model with an attention mechanism, which generates contextualized textual utterance representations by attending to past utterance representations in a given window. The process of generating utterance representation

from token vectors can be summarized as follows:

$$u_i^t = \text{maxpool}(\text{CNN}([w_i^{1,t}, w_i^{2,t}, \ldots, w_i^{k,t}]))$$

$$(14)$$

$$c_i^t = \sum_w \gamma_i \text{LSTM}(u_i^t) \in \mathbb{R}^{d_h} \qquad (15)$$

Similarly, we compute a contextualized audio representation $c_i^a$ for each utterance and pass both representations to the final late-stage fusion classifier module.

While the early-stage fusion encoder is structured with a Bi-LSTM and attention-based design, the unimodal encoder is designed using convolution and maxpooling to model utterances based on token or frame embeddings. The 1D convolution essentially functions as a temporal convolution, facilitating the integration of local context when generating utterance embeddings. Our experimental findings have demonstrated that the alternative encoder design, featuring convolutional filters, outperforms the use of a similar LSTM-based architecture in the unimodal-encoder branches. Our hypothesis is that the convolutional filters employed in the unimodal branches capture unique utterance representations compared to the LSTMs used in early-stage fusion encoders, resulting in this performance improvement.

### 3.3.3 Late-stage Fusion Classifier

This module integrates the multimodal and unimodal contextualized representations from the preceding layers by concatenating them, thereby producing a combined representation, which is subsequently fed into a two-layer MLP to perform DA classification.

$$z^i = \text{concat}(c_i, c_i^t, c_i^a) \qquad (16)$$
$$z^i = \text{ReLU}(W_{c1}z^i + b_{c1}) \qquad (17)$$
$$\hat{y}^i = \text{argmax}(\text{softmax}(W_{c2}z^i + b_{c2})) \qquad (18)$$

Here, $W_{\{c1,c2\}}$, and $b_{\{c1,c2\}}$ represent the weights and biases of the MLP layers in the classifier, respectively. The final MLP layer projects the joint representation to a $C$-dimensional embedding space, where $C$ is the number of classes. Softmax activation is applied to calculate the probability of the utterance belonging to each class. Finally, the argmax operation is applied to predict the DA corresponding to the utterance.

| Dataset | $|C|$ | $|l|$ | Dialogs | | | Utterances | | |
|---------|------|------|-------|-----|------|-------|------|------|
| | | | Train | Val | Test | Train | Val | Test |
| MRDA | 52 | 1442.5 | 51 | 12 | 12 | 75K | 16.4K | 16.7K |
| EMOTyDA | 12 | 9.6 | 727 | 104 | 208 | 6.9K | 1K | 2.1K |

Table 2: Number of classes $|C|$, utterances per dialog $|l|$, and number of dialogs and utterances in each split.

## 4 Experimental Setup

### 4.1 Datasets

We conduct experiments and present our findings based on two publicly available benchmark datasets that contain audio recordings of multi-party conversations: the Meeting Recorder Dialog Act (MRDA) corpus (Shriberg et al., 2004) and the EMOTyDA dataset (Saha et al., 2020). We provide the statistics of the datasets in Table 2.

**MRDA** The MRDA corpus consists of 75 multi-party meetings, each of which is considered a single dialog. The dialogs are on average of 1442.5 utterances in length. The dataset provides both oracle transcriptions and corresponding audio signals, enabling online DA classification. We split the meetings into 51 train, 12 validation and 12 test meetings. The MRDA corpus adopts a DA labeling scheme comprised of 52 distinct DA tags (Dhillon et al., 2004) that can be further clustered at different levels of granularity.

Specifically, these dialog acts are grouped into a general set of 12 tags and a basic set of 5 tags. The "basic-tags" grouping scheme introduced by Ang et al. (2005) has been widely adopted by subsequent studies, including Ortega and Vu (2017); Li et al. (2019); Raheja and Tetreault (2019); He et al. (2021). However, we contend that relying solely on the 5 basic tags limits our ability to capture the nuanced roles of utterances in the dialog. Hence, we use the full set of DA tags for a more fine-grained analysis.[1]

**EMOTyDA** The EMOTYDA dataset contains DA annotations for two popular datasets for emotion recognition in conversations: MELD (Poria et al., 2019) and IEMOCAP (Busso et al., 2008). For consistency with the choice of our first dataset, we choose the multiparty split of the EMOTyDA data which is the DA adaptation of the MELD corpus. We refer to this split as EMOTyDA in the rest of the

paper. The dataset consists of 1039 short dialogs with an average length of 9.6 utterances. The 12 most commonly occurring DA tags out of the 42 SWBD-DAMSL (Jurafsky and Shriberg, 1997) tags were used to annotate utterances of the EMOTyDA dataset. Saha et al. (2020) argue that, given the short nature of the dialogs, 12 tags are enough to capture the granular roles of the utterances in the dialogs, and use 831 dialogs for training and 208 dialogs for testing. We further split the original training set into 727 training and 104 validation dialogs and kept the original test set intact for a fair comparison.

### 4.2 Evaluation Metric

Both the MRDA and the EMOTyDA datasets contain a high degree of class imbalance, as evident from the label distributions presented in Appendix A. The problem is more severe in the MRDA dataset, where the five most frequent labels, namely *Statement (s), Backchannel (b), Floor Holder (fh), Acknowledgement (bk)*, and *Accept (aa)*, collectively account for approximately 65% of the total utterances. In contrast, the least frequent 12 tags constitute just 1% of the dataset. Similarly, in the EMOTyDA dataset, the top 5 classes, namely *Statement-NonOpinion (s), Question (q), Answer (ans), Statement-Opinion (o),* and *Others (oth)*, make up approximately 80% of the data.

Thus, we argue that while previous works have predominantly used accuracy as a metric for reporting model performance, it is not the most suitable metric for evaluating the performance of models trained on these imbalanced datasets. An accuracy-based evaluation can be misleading, as a model that performs well on the majority classes may achieve an overall higher accuracy while neglecting the minority classes. To address this limitation, we use the macro F1 score as our primary evaluation metric. This metric provides a more balanced assessment of model performance in this highly imbalanced setting. We also report the accuracy of the models to be consistent with prior work.

---

[1]Different granularities of DA labels are available here: https://github.com/NathanDuran/MRDA-Corpus

### 4.3 Implementation Details

We employ Huggingface [2] to utilize pre-trained RoBERTa and Whisper models for our feature extraction modules. The proposed models are trained using Categorical Cross-entropy loss function and optimized using the Adam optimizer. Appendix D provides detailed descriptions on hyperparameters, and Appendix E provides descriptions on compute hardware used and time requirements.

## 5 Results and Analysis

### 5.1 Baseline and Model Variants

We have evaluated our proposed **Hierarchical Fusion (HF) model** for online DA classification by comparing it against three strong related baselines and four variants of our own model.

#### 5.1.1 Model variants

**EF Model** consists solely of the Early-stage Fusion Encoder component of our proposed model. **Audio-Only Model** is a variant of our model that exclusively utilizes audio as input. In this configuration, the textual utterance encoder from the EF Model is removed. **Text-Only Model** is the text counterpart of the Audio-Only Model, using only text as input. **LF Model** is a late-stage fusion variant that leverages the Audio-Only and Text-Only models to generate contextualized unimodal representations. In this variant of the model, modality fusion occurs after contextualization while in the EF Model, the contextualization occurs after modality fusion. Finally the **HF Model** comprises of both the Early-stage Fusion Encoder and convolution based Unimodal Encoders.

#### 5.1.2 Related baselines

Related baselines we compare our model with are:

**Lexico-acoustic Model:** Ortega and Thang Vu (2018) is one of the very few related works that performs multimodal DA classification which creates an even ground for comparison with our proposed multimodal DA classification approach. **MulT:** Tsai et al. (2019) is originally proposed for multimodal emotion recognition which uses transformer architecture and cross-modal attention. We adapt the model for online DA classification due to its reproducibility and popularity as a relevant baseline in recent works. **Speaker Turn Modeling:** He et al.

---
[2] https://huggingface.co/

(2021) is the most recent baseline for DA classification . We adapt it for online DA classification by training it with only ASR generated transcriptions.

### 5.2 Audio Features

Various types of acoustic features have been explored in the literature for different audio processing tasks such as emotion recognition, sentiment analysis and DA classification. Mel Spectrograms and MFCC (Mel-Frequency Cepstral Coefficients) are most prominently used (Ortega and Thang Vu, 2018; He et al., 2018; Saha et al., 2020; Rejaibi et al., 2022; Chudasama et al., 2022). In recent years, the extended Geneva Minimalistic Acoustic Parameter Set (eGeMAPS), an expert knowledge-based compilation of 88 acoustic features (Li et al., 2020; Meng et al., 2022; Haider et al., 2021) and transformer-based acoustic feature extraction method using wav2vec2 (Pepino et al., 2021; Sharma, 2022) have gained attention for modeling emotions from audio signals. For our proposed approach, we extract the last hidden states of Whisper encoder as our acoustic features for input utterances.

| Audio Feature | Accuracy ↑ | Macro F1 ↑ |
|---|---|---|
| MFCC | 49.55 | 8.42 |
| eGeMAPs | 48.95 | 8.31 |
| Wav2Vec2 | 52.52 | 18.92 |
| Whisper Encoder | **55.63** | **25.44** |

Table 3: Performance of the Audio-Only model using various audio features on MRDA dataset.

We have evaluated the performance of our Audio-Only model on the MRDA dataset, employing the aforementioned feature sets. As shown in Table 3, audio features extracted from the Whisper encoder yield the highest performance in terms of both accuracy and F1. An improvement of ~6 percentage points is achieved over the nearest feature.

### 5.3 Online DA classification

Given that we are exploring a novel setup of online DA classification, which involves a fine-grained analysis of MRDA, we lack directly comparable results from existing systems. Therefore, we adapt three relevant baselines to suit the requirements of online DA classification. To ensure a fair comparison, we use Whisper-generated transcripts and Whisper encoder features for training and evaluating all the baseline models on the two datasets. The results in Table 4 indicate the promising performance improvements achieved by the proposed

| Model | MRDA | | EMOTyDA | |
|---|---|---|---|---|
| | Accuracy↑ | Macro F1↑ | Accuracy↑ | Macro F1↑ |
| Lexico-acoustic Model (Ortega and Thang Vu, 2018) | 59.71 | 25.52 | 54.88 | 38.72 |
| MulT (Tsai et al., 2019) | 58.18 | 24.06 | 47.22 | 36.68 |
| Speaker Turn Modeling (He et al., 2021) | 54.46 | 25.31 | 50.24 | 38.93 |
| HF Model | **59.86** | **29.11** | **55.58** | **41.95** |

Table 4: Performance comparison of the proposed and baseline models on MRDA and EMOTyDA datasets.

| Model | MRDA | | EMOTyDA | |
|---|---|---|---|---|
| | Accuracy↑ | Macro F1↑ | Accuracy↑ | Macro F1↑ |
| Audio-Only Model | 55.63 | 25.44 | 49.06 | 36.71 |
| Text-Only Model | 54.39 | 23.72 | 52.07 | 36.86 |
| LF Model | 55.66 | 25.78 | 50.82 | 37.44 |
| EF Model | 57.69 | 28.73 | 53.03 | 39.5 |
| EF Model + Cross Attention | 57.9 | 28.63 | 52.28 | 37.96 |
| HF Model | 59.86 | **29.11** | **55.58** | **41.95** |
| HF Model + Cross Attention | **60.26** | 28.9 | 55.25 | 41.4 |

Table 5: Ablation study of the proposed model.

model in online DA classification. Compared to the relevant baselines, our model demonstrates an increase in F1 score of ~3.5 percentage points on the MRDA dataset and ~3 percentage points on the EMOTyDA dataset.

Notably, while the other baseline models incorporate both audio signals and noisy transcriptions, the Speaker Turn Model stands out by achieving a higher F1 score for the EMOTyDA dataset, despite being trained only on the noisy transcriptions. This can be attributed to the fact that the ASR generated transcriptions on EMOTyDA are cleaner. Audio signals in the MRDA dataset are inherently noisy, susceptible to inter-channel interferences, leading to even noisier ASR transcriptions.

Overall, the similar performance gain on both the datasets using the proposed model indicates the robustness of our approach.

### 5.4 Ablation Study

To further analyze the impact of the individual components of our proposed model we perform a comprehensive ablation analysis (see Table 5). We compare the results of the complete HF model with the other variants described in subsubsection 5.1.1.

First, we examine the performance of the modality-specific models on both datasets. The Audio-Only model outperforms the Text-Only model on the MRDA dataset, while the reverse scenario is observed for the EMOTyDA dataset. This discrepancy can be attributed to the noisy na-

ture of the MRDA audio signals, which introduces inherent challenges in accurate transcription by the ASR system. Furthermore, the LF model demonstrated a minor improvement over the individual modalities in terms of F1 score for both datasets.

The most significant performance improvement was achieved by using early stage fusion (EF). The EF model exhibited almost 2 percentage points of improvement in F1 score over the LF model for both the datasets. This component addresses the major limitation of the previous DA classification research efforts which used multimodality at a later stage of dialog modeling. Finally, by incorporating the unimodal encoders into the EF model, we observe an increased overall accuracy of the model by approximately 2 percentage points without sacrificing the F1 score.

We do not observe any notable improvement of the model performance by incorporating cross-attention while utterance encoding. This observation is aligned with the findings of Rajan et al. (2022) for a similar task. Overall, we obtain approximately 5.5 F1 points and 4.5 F1 points improvement over the textual model for the MRDA and EMOTyDA datasets respectively.

### 5.5 Class-wise Performance

To understand the effectiveness of our multimodal HF model, We compare the performance of our HF model and Text-only model on individual classes. Out of 52 MRDA classes, We achieve a performance

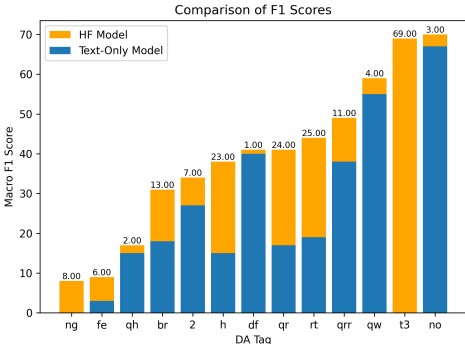

Figure 2: Impact of HF model on individual classes.

| Model | Multimodal | Accuracy |
|---|---|---|
| Ortega and Thang Vu (2018) | Y | 84.7 |
| Ravi and Kozareva (2018) | N | 86.7 |
| Raheja and Tetreault (2019) | N | 91.1 |
| He et al. (2021) | N | 91.4 |
| Colombo et al. (2020) | N | 91.6 |
| li et al. (2018) | N | 91.7 |
| Li et al. (2019) | N | 92.2 |
| Chapuis et al. (2020) | N | **92.4** |
| HF Model (online) | Y | 82.6 |
| HF Model (oracle) | Y | 91.8 |

Table 6: Comparison with state-of-the-art models on MRDA dataset trained with basic tags.

| Model | Accuracy | F1 Score |
|---|---|---|
| Saha et al. (2020) | | |
| DA only (Text + Audio) | 40.30 | 41.16 |
| DA + ER (Text + Audio) | 49.42 | 41.69 |
| DA + ER (Text + Video) | 51.00 | 44.52 |
| HF Model (online) | 55.58 | 41.95 |
| HF Model (oracle) | **63.42** | **50.3** |

Table 7: Comparison with state-of-the-art models on EMOTyDA dataset

gain for 32 classes. Among the 7 question classes, the HF model experiences a performance gain on 5 question classes with 24 F1 points improvement for *Or Question (qr)*. We can see the performance boost on a selected set of classes in Figure 2. For some classes like, *Negative Answer (ng), Third-party Talk (t3)* the Text-Only model fails to correctly classify any utterance, while HF model achieves an F1 score of 8 and 69 for these 2 classes. Similar trends have been observed for the EMOTyDA dataset, where the HF model achieves a performance gain on 10 out of 12 classes.

## 5.6 Comparison with SOTA Models

While our research primarily focuses on designing an efficient DA classification model for an online setup, we also measure the performance of our model using oracle transcriptions. This allows us to assess the effectiveness and robustness of our proposed approach in comparison to the state-of-the-art DA classification models. From Table 6, we can observe that, our proposed HF model outperforms all but two existing models when trained with oracle transcriptions. Notably, Li et al. (2019) achieves higher performance by jointly modeling DAs and topics, leveraging the complementary information from both tasks. From the ablation studies of the corresponding paper, it is evident that without employing the topic modeling jointly with DA classification, the model performance decreases by 2% and falls below the performance of our proposed model. Additionally, Chapuis et al. (2020) is a large model pre-trained on OpenSubtitles, a massive corpus of spoken conversations and fine-tuned on relevant datasets for DA classification. The superior performance can be attributed to the pre-training on relevant corpus and hence performs slightly better than our approach which is

only trained on DA classification dataset. We conduct similar experiments for the EMOTyDA dataset. The only state-of-the-art model for this dataset is Saha et al. (2020), which jointly models DA and emotion (ER). Our proposed model outperforms their best model, jointly trained on DA and ER with text and video, by 12% in terms of accuracy and 6 F1 points.

## 6 Conclusion

We have put forward a Hierarchical Fusion Model for online Dialogue Act (DA) classification. The early-stage fusion encoder component of the proposed model generates multimodal utterance representations more effectively. We also propose a novel audio feature extraction method using Whisper. Our detailed examination of different parts of our model confirms their relative importance and explains the effectiveness of our appraoch. The proposed Hierarchical Fusion Model yields a remarkable 3% improvement in the F1 score over the related baseline, and a noteworthy 5% improvement over using a text based model for online DA classification. Inspired by our exciting results from early fusion, in future we want to investigate even more granular fusion of the modalities at word level from unaligned data.

## Limitations

One of the major limitations that we face is the GPU memory restrictions. We employ two large language models RoBERTa and Whisper for feature extraction. However, we have not tuned the parameters of these models while extracting features. We believe an end-to-end training by incorporating these feature extraction models in the training will produce better features fordialog modeling. However, having a limited access to GPUs restrict us from doing so.

Another limitation that we have faced while working with the MRDA corpus is the inherent alignment issues of the dataset. For some of the meeting recordings, the audio signal was not perfectly aligned with the provided timestamps for utterances.

## Ethics Statement

Online DA classification plays a vital role in conversational agents and chatbots. The accurate identification and categorization of Dialogue Acts are vital for understanding the intentions, queries, and responses within a conversation. If the Dialogue Act recognition systems falter or generate unreliable outcomes, it can result in misunderstandings, misinterpretations, and flawed responses from the conversational agents. Hence, it is crucial to exercise caution when integrating Dialogue Act classification systems into real-world applications. A comprehensive evaluation of their performance, dependability, and resilience becomes essential before widespread deployment. Rigorous testing, validation, and ongoing improvement efforts should be undertaken to enhance their accuracy, adaptability, and applicability in real-world conversational contexts. By ensuring the quality and reliability of Dialogue Act recognition systems, we can mitigate the potential negative impacts on end users and elevate the overall user experience in interactions with conversational agents.

## Acknowledgements

We would like to thank the anonymous reviewers for their valuable feedback and input. Research was sponsored by the Army Research Office and was accomplished under Grant Number W911NF-20-1-0002. The views and conclusions contained in this document are those of the authors and should not be interpreted as representing the official policies, either expressed or implied, of the Army Research Office or the U.S. Government. The U.S. Government is authorized to reproduce and distribute reprints for Government purposes notwithstanding any copyright notation herein.

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

# A  Label Distribution for MRDA and EMOTyDA

Figure 3 shows the label distribution for the MRDA and EMOTyDA datasets.

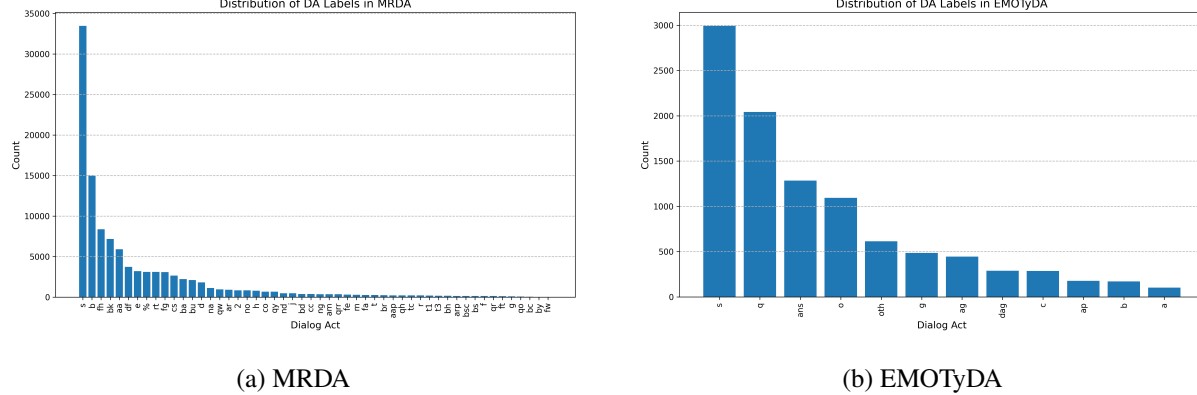

| (a) MRDA | (b) EMOTyDA |

Figure 3: Label distribution of the two datasets.

| Feature | Macro F1 | Accuracy |
|---|---|---|
| Last hidden state | **29.11** | **59.86** |
| Average of last 4 hidden states | 28.42 | 59.34 |

Table 8: Comparison of different configurations of RoBERTa features

| Name | Best Value |
|---|---|
| Hidden dimensions of encoders | 128 |
| Kernel size | 5 |
| No of channels for conv layers | 256 |
| Window size for context modeling | 3 |

Table 9: Choice of hyperparameters

## B  Configuration of Textual Feature Extraction

We use pre-trained RoBERTa as feature extractor to extract token embeddings from the textual input. The choice of this feature extractor is consistent with contemporary works on DA classification (He et al., 2021) and other NLP tasks (Zheng et al., 2023; Chudasama et al., 2022) in the literature. RoBERTa is also more recent and proven to be better than BERT on several NLP benchmarks and hence a better choice of feature extractor.

We experimented with two variants of RoBERTa features, one where the token embeddings are the average of last 4 hidden states and the other where the token embeddings are solely from last hidden state. As shown in Table 8, we did not find any significant improvement using the intermediate layers and hence continued with the last hidden states for feature extraction for all the experimentation.

## C  Configuration of Audio Feature Extraction

For extracting frame embeddings from the audio, we used the pre-trained model, Whisper and followed the same setup that is used in original Whisper implementation for pre-processing audio inputs. The input audio utterance is padded or clipped to 30s and a 25ms window with 10 ms stride is employed to extract the mel-spectrograms. The

choice of window size and stride for Whisper is consistent with that of other audio feature extraction methods across the literature. The final output from the whisper encoder is 1500 frames per utterance with 512 dimensional embedding per frame.

## D  Hyperparameters

To control the learning rate, we implemented a linear LR scheduler that gradually decreased the rate from the initial value of 1e-3 after 5 epochs. For the MRDA dataset, our experiments are conducted over 40 epochs, while for the EMOTyDA dataset, the experiments were carried out for 20 epochs. To prevent overfitting, we used early stopping, ceasing training if no improvement in validation macro F1 score was observed for 5 consecutive epochs.

The best model for inference on the test set was selected based on macro F1 score on the validation set. To ensure the reliability of the results, each experiment was repeated 5 times with different random seeds, and the reported results represent the average performance across these runs.

A list of hyperparameters and the chosen values upon experimentation is shown in Table 9.

## E  Training and Inference

All experiments were performed on a single NVIDIA RTX A6000 GPU. Training an epoch

on the MRDA dataset took approximately 40 minutes, with an additional 12 minutes each for validation and testing. Training time can be reduced by caching the audio features but it in turns increase the memory requirement on the CPU. Training an epoch on the EMOTyDA dataset took approximately 7 minutes, with an additional 2 minutes each for validation and testing. The latency of the model at inference time is 0.16 seconds to infer dialog act for each utterance.