# OpenReview forum: "Hierarchical Fusion for Online Multimodal Dialog Act Classification"
_EMNLP/2023/Conference — EMNLP 2023 Findings_

### Official Review · Reviewer_ybDR · 2023-08-04

**Soundness:** 4

**Excitement:**

3: Ambivalent: It has merits (e.g., it reports state-of-the-art results, the idea is nice), but there are key weaknesses (e.g., it describes incremental work), and it can significantly benefit from another round of revision. However, I won't object to accepting it if my co-reviewers champion it.

**Paper Topic And Main Contributions:**

This paper invesitages approach to dialog act recognition from textual and audio inputs.

The main contribution is the proposed modality-fusing architecture:
- Feature extraction
  - Extract textual token embeddings and audio frame embeddings from pretrained RoBERTa and Whisper models, respecitvely.
- Unimodal encoding
  - Compute unimodal utterance embeddings using convolution + max pooling.
  - Compute unimodal dialog embedding from contextualized unimodal utternace embeddings using LSTM + attention.
- Modality-fusing encoding (called early-stage fusion)
  - Compute unimodal utterance embedding using BiLSTM + attention.
  - Obtain multimodal utterance embedding by concatenating unimodal embeddings.
  - Compute multimodal dialog embedding from contexutalized multimodal utterance embeddings using BiLSTM + attention.
- Classification (called late-stage fusion)
  - Combine the unimodal dialog embeddings and multimodal embedding to compute the final output.

The proposed model is evaluated on the MRDA and EMOTyDA datasets.
- In the customized online setting (that uses ASR transcripts from Whisper as textual inputs), the proposed model outperforms several relatively weak baselines.
- In the oracle settings (that uses oracle texts as textual inputs), the proposed model slightly lags behind a SoTA unimodal baseline.
- The ablation study shows the existence of the modality-fusing component contributes the most to its performance.

In general,
- the paper proposed an effective method of fusing multimodal features for dialog act recognition;
- the evaluation results are not particularly strong in comparison with unimodal models from before 2021 (see Q1 below);
- it is not very clear whether **modality fusion** plays an important role in the proposed model (see Q2 below).

**Questions For The Authors:**

Q1: Why not compare with SoTA model from Section 5.6 in the online setting experiments (Section 5.3)?

Q2: The utterance encoder designs in the unimodal encoder and early-stage fusion encoder are different. The former is conv + max pooling, and the latter is BiLSTM + attention. What is the reason for making such decision? Due to the existence of alternative unimodal utterance encoders in the early-stage fusion encoder, we can conclude that **the early-stage fusion encoder** contributes the most, but it is questionable how much **modality fusing** contributes.

Q3: Frame embedding sequences are usually very long and hard to encode without compressing into short sequences. The early-stage fusion encoder uses utterance BiLSTM to encode audio embeddings at a frame level. Will not it be very memory consuming and difficult to train? What is the average length of frame embedding sequences in your experiment?

Q4: In the last paragraph of Section 3.3.1, the authors say cross attention is applied to textual and audio representations, but details of the cross attention are neither presented in Figure 1 nor by any equations. Can the authors provide more details?

**Reasons To Accept:**

RA1: An effective method of combining textual and audio features for dialog act recognition (or similar tasks).

RA2: Confirmed performance improvement by incorporating the proposed unimodal and modalitiy-fusing components.

**Reasons To Reject:**

RR1: The experiment design and model architecture design prevent readers from concluding what factor is the most important for improving task performance (see Q1, Q2 below).

RR2: Important technical details are missing (see Q4 below).

**Reproducibility:**

1: Could not reproduce the results here no matter how hard they tried.

**Reviewer Confidence:**

4: Quite sure. I tried to check the important points carefully. It's unlikely, though conceivable, that I missed something that should affect my ratings.

**Typos Grammar Style And Presentation Improvements:**

It is better to put Section 5.2 after the main results (i.e. at least after the current Section 5.4).

---

> ### Author Rebuttal · Authors · 2023-08-29
>
> Thanks you for your valuable feedback and suggestions. We respond to your feedback and comments below.
>
> ### Comparison with SoTA
>
> > In the oracle settings (that uses oracle texts as textual inputs), the proposed model slightly lags behind a SoTA unimodal baseline.
>
> **Response:** In the oracle setup where we have used clean transcriptions to train and evaluate the model, our proposed model outperforms all but two of the related baselines. These results are justified in the *section 5.6, lines 582--589* of the paper. To elaborate, one of the two unimodal approaches are Li et al. [1], which joint models DA and topic to enhance the performance of DA classification. From the ablation studies of the corresponding paper, it is evident that without employing the topic modeling task jointly with DA classification, the model performance decreases by 2% and falls below the performance of our proposed model. The second one, Chapius et al. [2] is essentially a large model pre-trained on OpenSubtitles, a massive corpus of spoken conversations and fine-tuned on relevant datasets for DA classification. The superior performance can be attributed to the pre-training on relevant corpus and hence performs slightly better than our approach which is only trained on DA classification dataset.
>
>
> ### Doubts regarding baselines
> > In the customized online setting (that uses ASR transcripts from Whisper as textual inputs), the proposed model outperforms several relatively weak baselines.
>
> > The evaluation results are not particularly strong in comparison with unimodal models from before 2021 (see Q1 below).
>
> > Q1. Why not compare with SoTA model from Section 5.6 in the online setting experiments (Section 5.3)?
>
> **Response:** We respectfully differ with the reviewers notion that the baselines are weak. We have chosen 3 baselines and we have justified the choice of our baselines in the paper (lines 450 - 460). The first paper we have chosen as the baseline is the **only** related work that performs multimodal DA classification which creates an even ground for comparison with our multimodal DA classification approach. The second baseline is a Multimodal Transformer based approach which is one of the most cited (765) papers in the field of multimodal sentiment analysis/emotion recognition and cited as a relevant baseline in the recent papers. We have chosen this paper as the code is publicly available and hence it is easily reproducible. The third baseline is the one of the most recent works published on DA classification. For EMOTyDA, we only have one baseline paper and our paper outperforms it by a large margin.
>
> The SoTA model, Chapius et al. 2021 [2] in the table 5.6 is a large model pre-trained on dialog specific corpus, OpenSubtitles and fine-tuned on relevant datasets for DA classification. The model is not publicly available and considering the scale of its training and development, implementing the work and comparing the results with our work is beyond our capacity. However, we have compared our work with 2 of the other relevant works (which were publicly available) presented in table 5.6, Ortega & Thang Vu (2018) [3] and He et al. (2021) [4], as relevant baselines in the online setup. They are referred to as **Lexico-acoustic Model** and **Speaker Turn Modeling** in the table 5.3 respectively and from the performance comparison, it is evident that our proposed model outperforms the two models by quite a good margin.
>
> ### Role of modality fusion
>
> > It is not very clear whether modality fusion plays an important role in the proposed model (see Q2 below).
> > Q2: The utterance encoder designs in the unimodal encoder and early-stage fusion encoder are different. The former is conv + max pooling, and the latter is BiLSTM + attention. What is the reason for making such decision? Due to the existence of alternative unimodal utterance encoders in the early-stage fusion encoder, we can conclude that the early-stage fusion encoder contributes the most, but it is questionable how much modality fusing contributes.
>
> **Response:** "Indeed, the two architectures used for unimodal feature extraction in the early-stage fusion encoder and the unimodal encoders are different. The early-stage fusion encoder has Bi-LSTM+attention based design which is a more straightforward choice based on the literature to model the utterances from token embeddings. The conv+maxpooling architecture is also employed in the unimodal encoders to model the utterances from token or frame embeddings. 1D convolution essentially works as a temporal convolution that helps to incorporate local context while computing utterance embedding from token embeddings. Our experiments show that using an alternative design of encoder with conv filters in the unimodal-encoder branches provide better results than using the similar LSTM based architecture. Essentially, this was a design choice based on empirical results. Our hypothesis is, the conv filters in the unimodal branches learn different representations of utterances than LSTMs employed in the early-stage fusion encoders and hence contribute to the better results.
>
> We are unclear on what is meant by the sentence 'Due to the existence of alternative unimodal utterance encoders in the early-stage fusion encoder, we can conclude that the early-stage fusion encoder contributes the most, but it is questionable how much modality fusing contributes,' since earlier in the review, the reviewer seems to equate early-stage fusion with modality fusing: `Modality-fusing encoding (called early-stage fusion)`. It would be helpful to get a bit more clarification on this point in order to write a response but we are trying to explain the confusion to the best of our understanding of the comment.
>
> We are under the impression that, the reviewer is convinced by our early-stage fusion encoder component but not entirely confident about how much the early fusion of the modalities is contributing to the performance gain and the overall performance gain could be attributed to the alternative designs of encoders rather than fusion. To validate the importance of fusing the modalities early, such as in the early-stage fusion encoder, we employ an alternative design, where we have the 4 unimodal utterance encoders (2 text based and 2 audio based) and we concatenate the representations from the 4 encoders to further use them in the classifier.
>
> |  Experiment   | Macro F1|    Accuracy  |
> | :---        |    :----:   |          ---: |
> | Original HF model    | **29.11**       |  **59.86** |
> | Alternative design without early fusion  | 27.29      | 56.05      |
>
> We can observe that, the alternative design yields almost 2% less F1 score and almost 4% less accuracy compared to our HF model. This results reinforce the importance of modality fusion at the early stage of the model and context modelling. We not only have designed utterance encoders to extract good embeddings, we have carefully designed the model with fusion components that contribute to the overall results.
>
> ### Clarification regarding frame embedding sequences
> > Q3: Frame embedding sequences are usually very long and hard to encode without compressing into short sequences. The early-stage fusion encoder uses utterance BiLSTM to encode audio embeddings at a frame level. Will not it be very memory consuming and difficult to train? What is the average length of frame embedding sequences in your experiment?
>
> **Response:** For extracting frame embeddings, we used the pre-trained model, Whisper and followed the same setup that is used in Whisper. The input audio utterance is padded or clipped to 30s and a 25ms window with 10 ms stride is employed to extract the mel-spectrograms. The choice of window size and stride for Whisper is consistent with that of other audio feature extraction methods across the literature. The final output from the whisper encoder is 1500 frames per utterance with 512 dimensional embedding per frame. We have successfully trained our model using these features and details of training and GPU requirement is provided in appendix B and C of our paper.
>
> ### Clarification regarding cross-attention
> > Q4: In the last paragraph of Section 3.3.1, the authors say cross attention is applied to textual and audio representations, but details of the cross attention are neither presented in Figure 1 nor by any equations. Can the authors provide more details?
>
> **Response:** We have provided the details about implementation of cross-attention in lines 281-292. However, as we did not find notable performance improvement using cross-attention as mentioned in the paper (*section 5.4, lines 546-550*), we chose to not include the cross-attention component in the figure. The details of cross-attention implementation is provided below and we also plan to incorporate these details in the final paper.
>
> Following the conventions of the paper, we use equation 3 from the paper to compute the token/frame representations from Bi-LSTM, $ h_{i}^{j, t} $ and $ h_{i}^{j, a} $ from text and audio modalities. Now while calculating cross attention, we use either the text modality as source and audio as target or vice versa. We want to compute two cross-attention based utterance embedding, one is $s_{i}^{t,a}$ where, the attention weights are calculated using the dot product of the two modality representations and based on the attention weights the token embeddings are aggregated to an utterance embedding and on the other hand, another is $s_{i}^{a,t}$ where, the attention weights are calculated using the dot product of the two modality representations and based on the attention weights the frame embeddings are aggregated to an utterance embedding. To achieve this, we first pass the token embeddings and frame embeddings through a simple MLP layer followed by a tanh activation to generate intermediate token/frame representations.
>
> $$ p_{i}^{j, t} = \tanh(W_1 h_{i}^{j, t} + b_1)$$
> $$ p_{i}^{j, a} = \tanh(W_2 h_{i}^{j, a} + b_2)$$
>
> Now we use, matrix multiplication to compute pairwise dot-products between intermediate token and frame representations. If $P_{i}^{t}$ and  $P_{i}^{a}$ are the two matrices of all the intermediate token and frame representations in the utterance, we compute the following matrix multiplication to compute the attention weights. Here the dimension of  $P_{i}^{t}$ is  (num_tokens $ \times $ embedding_dim) and the dimension of $P_{i}^{a}$ is (num_frames $\times$ embedding_dim).
>
> $$ S = P_{i}^{t} \times (P_{i}^{a})^{T} $$
>
> The dimension of this matrix is (num_tokens $\times$ num_frames). Then we use softmax to normalize the attention weights. Now we apply another MLP transformation on the audio frame embedding matrix and finally multiply it with the attention weights to get the attended token embeddings,
>
> $$ P_{i}^{t'} = S \times MLP(P_{i}^{a})$$
>
> The dimension of this matrix is (num_tokens $\times$ embedding_dim). Finally, we average all the attended-token embeddings to generate the utterance embedding, $s_{i}^{t,a}$. In a similar way we can compute the  $s_{i}^{a,t}$. The final utterance embedding will be the concatenation of the self and cross attended utterance embeddings.
>
> $$ s_{i} = concat(s_{i}^{t}, s_{i}^{a}, s_{i}^{t,a}, s_{i}^{a,t})$$
>
> The rest of the model remains unchanged.
>
> - [1] Li et al. 2019.  [A dual-attention hierarchical recurrent neural network for dialogue act classification](https://aclanthology.org/K19-1036/).
>
> - [2] Chapius et al. 2021. [Hierarchical pre-training for sequence labelling in spoken dialog](https://aclanthology.org/2020.findings-emnlp.239/).
>
> - [3] Ortega and Thang Vu 2018. [Lexico-Acoustic Neural-Based Models for Dialog Act Classification](https://ieeexplore.ieee.org/document/8461371).
>
> - [4] He et al. 2021. [Speaker Turn Modeling for Dialogue Act Classification](https://aclanthology.org/2021.findings-emnlp.185/).

---

### Official Review · Reviewer_YaT8 · 2023-08-05

**Typos Grammar Style And Presentation Improvements:** 1. The author proposed "Hierarchical …
**Soundness:** 3

**Excitement:**

3: Ambivalent: It has merits (e.g., it reports state-of-the-art results, the idea is nice), but there are key weaknesses (e.g., it describes incremental work), and it can significantly benefit from another round of revision. However, I won't object to accepting it if my co-reviewers champion it.

**Paper Topic And Main Contributions:**

This paper present an multimodal approach that employs a hierarchical fusion technique to early-stage combine two modalities, yielding remarkable performance surpassing that of utilizing individual modalities for online DA classification. The proposed approach adeptly incorporates multimodal fusion strategies, effectively capturing both utterances and their context. The author also introduced a audio feature extraction process by using the pre-trained Whisper model to enhance the modeling of dialogs based on audio, resulting in improved DA classification performance. The result showed the proposed system achieved 3% improvement in the F1 score compared to the related baseline and an impressive 5% improvement over using a text-based model for online DA classification on MRDA and EMOTyDA dataset.

**Questions For The Authors:**

Q.A: Any specific reason to use RoBERTa as the feature extractor for text rather than other foundation models?

Q.B: A lot of previous research showed using the intermediate layer outputs may lead better generalization compared to the last layer. Did author compare the features extracted by other layers?

Q.C: For the early stage fusion, can we also consider the positional encoding + Transformer as the fusion architecture? Is there any specific reason to use Bi-LSTM rather than other architectures?

Q.D: Typically, 1D Convolution model are used to extract the features from raw data such as word-level vectors and raw audio wav. Considering the proposal system have already used pre-trained model to generate the feature-level representations, why should we still use Conv models for the uni-modality encoders?

Q.E: Since Whisper also targets for the ASR task, just wondering if we can directly use the Whisper ASR transcripts for DA task? What about the performance compared to the proposed multimodal design?

Q.F:  Do we also need to consider the latency for online DA? Since it will be a real-time system, I think latency should be an important evaluation metric.

**Reasons To Accept:**

1. The paper presents a thorough investigation with comprehensive experiments and quantitative analysis, comparing both baseline models and ablations.

2. The results indicate that the introduced system outperforms state-of-the-art baselines, while also highlighting the effectiveness of the proposed hierarchical fusion design.

**Reasons To Reject:**

1. The novelty of hierarchical fusion is relatively limited, as the idea of combining uni-modality and multi-modality fusion is not new. Similar concepts have been previously introduced in dialog-based multimodal systems, and specific architectural designs require further explanation.

**Reproducibility:**

4: Could mostly reproduce the results, but there may be some variation because of sample variance or minor variations in their interpretation of the protocol or method.

**Reviewer Confidence:**

4: Quite sure. I tried to check the important points carefully. It's unlikely, though conceivable, that I missed something that should affect my ratings.

---

> ### Author Rebuttal · Authors · 2023-08-29
>
> Thank you for your valuable feedback and suggestions. We are glad that you perceive our experiments as thorough and comprehensive, and find our results compelling. We
> respond to your feedback and comments below.
>
> ### Comments
>
> > The novelty of hierarchical fusion is relatively limited, as the idea of combining uni-modality and multi-modality fusion is not new. Similar concepts have been previously introduced in dialog-based multimodal systems, and specific architectural designs require further explanation.
>
> We agree with the reviewer that the term 'Hierarchical Fusion' is used across the literature for multimodal fusion. However, in most of the recent literatures, the term hierarchical fusion is used in a different perspective compared to ours. For instance, in the papers Huang et al. [1], Majumder et al. [2] and Wang et al. [3], hierarchical fusion is used to represent a common idea where the features from two out three available modalities (audio, text and video) are concatenated first to synthesize bi-modal features. These bi-modal features are in turns concatenated to synthesize the tri-modal features which are eventually used for classification. On the contrary, in our proposed approach, we only have two features (text and audio). In **early-stage fusion**, we first extract the utterance embeddings individually from the modalities to combine them in a single utterance representation and then we incorporate contextual information. Whereas, in the unimodal encoder branches, we contextualize the encodings from the individual modalities separately. Finally we combine the three contextualized encodings, which is referred to as **late-stage fusion**. Essentially, in the context of our work, hierarchical fusion (HF) refers to fusing the two modalities before and after contextualization while the existing multimodal approaches at least for DA classification only fuse after contextualizing the features. From ablation studies presented in table 5, we observe that, we only get <1% improvement in F1 score if we fuse the modality representations after contextualization (LF model), while if we first combine and then contextualize (EF model), we get 3% improvement in F1 score. If we choose to combine at both levels (HF model), we get further improvement in F1 score and 2% improvement in accuracy.
>
> ### Questions
>
> > Q.A: Any specific reason to use RoBERTa as the feature extractor for text rather than other foundation models?
>
> **Response:** We use RoBERTa in order to be consistent with contemporary works on DA classification (He et al. [4]) and other NLP tasks (Zheng et al. [5], Chudasama et al. [6] etc.) in the literature. RoBERTa is also more recent and proven to be better than BERT on several NLP benchmarks and hence a better choice of feature extractor. In any case, the choice of the textual feature extractor is not the main contribution of this paper contribution, and thus we are fairly agnostic to it.
>
> > Q.B: A lot of previous research showed using the intermediate layer outputs may lead better generalization compared to the last layer. Did author compare the features extracted by other layers?
>
> **Response:** In the early stages of our experiment we employed both variants of RoBERTa features, one where the token embeddings are the average of last 4 hidden states and the other where the token embeddings are from last hidden state. As shown in the following table, we did not find any significant imrovement using the intermediate layers and hence continued with the last hidden states for feature extraction.
>
> |   Feature   | Macro F1|    Accuracy  |
> | :---        |    :----:   |          ---: |
> | Last hidden state      | **29.11**       |  **59.86** |
> | Average of last 4 hidden states  | 28.42       | 59.34      |
>
> > Q.C: For the early stage fusion, can we also consider the positional encoding + Transformer as the fusion architecture? Is there any specific reason to use Bi-LSTM rather than other architectures?
>
> **Response:** In the proposed model, LSTM + attention architechture is used for two different functionalities. The first one is modelling utterances from token (text) or frame (audio) embeddings and the second one is context modelling.
>
> For modelling utterances from token or frame embeddings, we are already extracting features from RoBERTa and Whisper models which use Transformer architecture to compute features. Essentially, we are using the LSTM+attention model on top of the features to aggregate the token embeddings into utterance embedding. We have also tried to experiment with fully self-attention based architecture for modelling utterances but that did not outperform the LSTM based encoder. We have experimented with different combinations of up-to 3 layers and 6 attention heads due to computational resource constraints.
>
> On the other hand, to model contexts, we do not need infinitely long context window because for modelling context-aware utterance embeddings, only previous few utterances are important and a simple LSTM model is sufficient to achieve that.
>
> Infact, one of the related works (Multimodal Transformer, MulT) that we have mentioned in the paper, uses end-to-end Transformer architecture and could not outperform our model, which indicates the importance of separate utternace and dialog modelling units we employ in the proposed model.
>
> > Q.D: Typically, 1D Convolution model are used to extract the features from raw data such as word-level vectors and raw audio wav. Considering the proposal system have already used pre-trained model to generate the feature-level representations, why should we still use Conv models for the uni-modality encoders?
>
> **Response:** Indeed, the 1d conv+maxpooling architecture is employed in the unimodal encoders to model the utterances from token or frame embeddings. 1D convolution essentially works as a temporal convolution that helps to incorporate local context while computing utterance embedding from token embeddings. We have used LSTM+attention for utterance modelling in the early-stage fusion encoder already and our experiments show that using an alternative design of encoder with conv filters provide better results than using the similar LSTM based architecture in the unimodal encoder branches. Essentially, this was a design choice based on empirical results. Our hypothesis is, the conv filters in the unimodal branches learn different representations of utterances than LSTMs employed in the early-stage fusion encoders and hence contribute to the better results.
>
> > Q.E: Since Whisper also targets for the ASR task, just wondering if we can directly use the Whisper ASR transcripts for DA task? What about the performance compared to the proposed multimodal design?
>
> **Response:** This is already implemented in the paper and in-fact we use Whisper to generate transcriptions for all the text-based experiments in the online setup of our work. The comparison of the Whisper generated text-only model with the HF model can be found in table 4 of the paper.
>
> > Q.F: Do we also need to consider the latency for online DA? Since it will be a real-time system, I think latency should be an important evaluation metric.
>
> **Response:** We agree with the reviewer that latency is a very important factor for online model. However, since our proposed model is the first of the kind, there is no baseline to compare the latency with. We have computed the latency of the model and it is 0.16 seconds to infer dialog act for one utterance, which will serve as a comparable baseline for future online DA classification works.
>
> - [1] Huang et al. 2023.  [Multimodal Sentiment Analysis in Realistic Environments Based on Cross-Modal Hierarchical Fusion Network](https://www.mdpi.com/2079-9292/12/16/3504).
>
> - [2] Majumder et al. 2018. [Multimodal sentiment analysis using hierarchical fusion with context modeling](sciencedirect.com/science/article/abs/pii/S0950705118303897).
>
> - [3] Wang et al. 2021.  [Hierarchical Multimodal Fusion Network with Dynamic Multi-task Learning](https://ieeexplore.ieee.org/document/9599141).
>
> - [4] He et al. 2021. [Speaker Turn Modeling for Dialogue Act Classification](https://aclanthology.org/2021.findings-emnlp.185/).
>
> - [5] Zheng et al. 2013.  [A Facial Expression-Aware Multimodal Multi-task Learning Framework for Emotion Recognition in Multi-party Conversations](https://aclanthology.org/2023.acl-long.861/).
>
> - [6] Chudasama et al. 2022. [M2FNet: Multi-modal Fusion Network for Emotion Recognition in Conversation](https://ieeexplore.ieee.org/document/9857351).

---

### Official Review · Reviewer_7qHe · 2023-08-08

**Typos Grammar Style And Presentation Improvements:** (573) Something is missing after "for…
**Soundness:** 4

**Excitement:**

4: Strong: This paper deepens the understanding of some phenomenon or lowers the barriers to an existing research direction.

**Missing References:**

The related work on dialog act recognition in the paper starts in 2017. Considering that starting date, the authors miss several papers related to the task. For instance, and non-exhaustively:

Bothe et al. (2018) A Context-based Approach for Dialogue Act Recognition using Simple Recurrent Neural Networks
Chen et al. (2018) Dialogue Act Recognition via CRF-Attentive Structured Network
Kumar et al. (2018) Dialogue Act Sequence Labeling Using Hierarchical Encoder With CRF
Wan et al. (2018) Improved Dynamic Memory Network for Dialogue Act Classification with Adversarial Training
Ribeiro et al. (2019) Deep Dialog Act Recognition using Multiple Token, Segment, and Context Information Representations
Ren et al. (2020) Intention Detection Based on Siamese Neural Network With Triplet Loss
Wang et al. (2020) Integrating User History into Heterogeneous Graph for Dialogue Act Recognition
Wang et al. (2021) Balance the Labels: Hierarchical Label Structured Network for Dialogue Act Recognition
Zelasko et al. (2021) What Helps Transformers Recognize Conversational Structure? Importance of Context, Punctuation, and Labels in Dialog Act Recognition

I'm not saying that all of these papers should be cited, but the authors should at least make it clear that the papers cited in the related work are just some examples of existing approaches to the task. However, I think that these papers should be checked, as they may provide insight for future work by the authors.

Furthermore, when possible, peer-reviewed publications should be cited instead of their pre-print counterparts. For instance, (Colombo, 2020) was published in the proceedings of AAAI.

**Paper Topic And Main Contributions:**

This paper presents a multimodal approach to dialog act recognition that combines audio and textual information and avoids the need for oracle transcriptions. The results on two datasets (MRDA and EMOTyDA) reveal competitive performance with state-of-the-art approaches.

NOTE: This is an emergency review, so I'm sorry for not going into much detail.

**Reasons To Accept:**

When considering real-time analysis of spoken interaction, the ability to work with automatic error-prone transcriptions is particularly important. This paper addresses that problem in the context of the dialog act recognition task, which is an important step in the understanding process.

**Reasons To Reject:**

When comparing the results with those of previous approaches on the MRDA corpus, the proposed approach is still outperformed by an unimodal approach, even when relying on oracle transcriptions. This suggests that the parts of the system that deal with textual information can be improved. Furthermore, the related work on dialog act recognition is missing several references.

**Reproducibility:**

4: Could mostly reproduce the results, but there may be some variation because of sample variance or minor variations in their interpretation of the protocol or method.

**Reviewer Confidence:**

4: Quite sure. I tried to check the important points carefully. It's unlikely, though conceivable, that I missed something that should affect my ratings.

---

> ### Author Rebuttal · Authors · 2023-08-29
>
> Thank you for your insightful feedback and suggestions! We feel encouraged by your appreciation towards our Dialog Act (DA) classification model, which is capable of working with automatic error-prone transcriptions. We
> respond to some of your feedback and comments below.
>
> ### Clarifying the results in the oracle setup
>
> > When comparing the results with those of previous approaches on the MRDA corpus, the proposed approach is still outperformed by an unimodal approach, even when relying on oracle transcriptions. This suggests that the parts of the system that deal with textual information can be improved.
>
> In the oracle setup where we have used clean transcriptions to train and evaluate the model, our proposed model outperforms all but two of the related baselines. These results are justified in the *section 5.6, lines 582--589* of the paper. To elaborate, one of the two unimodal approaches are Li et al. [1], which joint models DA and topic to enhance the performance of DA classification. From the ablation studies of the corresponding paper, it is evident that without employing the topic modeling task jointly with DA classification, the model performance decreases by 2% and falls below the performance of our proposed model. The second one, Chapius et al. [2] is essentially a large model pre-trained on OpenSubtitles, a massive corpus of spoken conversations and fine-tuned on relevant datasets for DA classification. The superior performance can be attributed to the pre-training on relevant corpus and hence performs slightly better than our approach which is only trained on DA classification dataset.
>
> ### Missing References
>
> > The related work on dialog act recognition in the paper starts in 2017. Considering that starting date, the authors miss several papers related to the task.
>
> We want to thank you for drawing our attention to all the relevant papers. Bothe et al. (2018), Chen et al. (2018), Kumar et al. (2018), Wang et al. (2021), and Zelasko et al. (2021) were actually explored during the literature review phases of the work but later missed in the paper. We plan to cite these and some of the other suggested papers in the final version of the paper.
>
> - [1] Li et al. 2019.  [A dual-attention hierarchical recurrent neural network for dialogue act classification](https://aclanthology.org/K19-1036/).
>
> - [2] Chapius et al. 2021. [Hierarchical pre-training for sequence labelling in spoken dialog](https://aclanthology.org/2020.findings-emnlp.239/).

---

### Meta-Review · Area_Chair_wRWz · 2023-09-20

**Recommendation:** 3

**Metareview:**

The paper addresses the task of multimodal dialog act classification in an online setting – raw audio and ASR text.

Reviewers appreciated the alignment of assumptions towards real-world applications. They also found the experimentation to be comprehensive and sound.

From a technical perspective, the excitement was average as the components proposed in the paper are fairly standard and known.

In addition, some of the choices and assumptions in the architecture were found to be confusing – which warrants improvements in the writing.

---

### Decision · Program_Chairs · 2023-10-07

**Decision:**

Accept-Findings

**Comment:**

The paper addresses the task of multimodal dialog act classification in an online setting – raw audio and ASR text.

Reviewers appreciated the alignment of assumptions towards real-world applications. They also found the experimentation to be comprehensive and sound.

From a technical perspective, the excitement was average as the components proposed in the paper are fairly standard and known.

In addition, some of the choices and assumptions in the architecture were found to be confusing – which warrants improvements in the writing.